# NF-κB–Dependent Snail Expression Promotes Epithelial–Mesenchymal Transition in Mastitis

**DOI:** 10.3390/ani11123422

**Published:** 2021-12-01

**Authors:** Haokun Liu, Ying Zhao, Yanfang Wu, Yutong Yan, Xiaoe Zhao, Qiang Wei, Baohua Ma

**Affiliations:** 1College of Veterinary Medicine, Northwest A&F University, Yangling, Xianyang 712100, China; lhkxn@nwafu.edu.cn (H.L.); yingzhao_educn@163.com (Y.Z.); wyf18829353119@163.com (Y.W.); xnytyan@163.com (Y.Y.); zhaoxe@nwsuaf.edu.cn (X.Z.); 2Key Laboratory of Animal Biotechnology, Ministry of Agriculture and Rural Affairs, Northwest A&F University, Yangling, Xianyang 712100, China

**Keywords:** NF-κB, Snail, EMT, mastitis, lipopolysaccharide, goat

## Abstract

**Simple Summary:**

Mastitis is a common and important clinical disease in ruminants, resulting in decreased milk production, infertility and delayed conception. If not treated promptly, mastitis may result in fibrotic mastitis. Although epithelial–mesenchymal transition (EMT) is a typical characteristic of fibrotic diseases, the relationship between EMT and mastitis remains largely unknown. NF-κB and Snail are key regulators of the EMT. In the present study, we found that lipopolysaccharide (LPS) induced EMT in primary goat mammary epithelial cells (GMECs). Additionally, the expression of Snail was induced by LPS and was inhibited by the suppression of the TLR4/NF-κB signaling pathway. The knockdown of Snail alleviated LPS-induced EMT and altered the expression of inflammatory cytokines. Finally, we found that the expression of key molecules of the TLR4/NF-κB/Snail signaling pathway was increased in mastitic tissues. This study provides evidence that LPS induces EMT in GMECs through the TLR4/NF-κB/Snail signaling pathway and lays a theoretical foundation for further exploration of the pathological mechanism and treatment of mastitis.

**Abstract:**

Mastitis is a common and important clinical disease in ruminants. This may be associated with inflammatory fibrosis if not treated promptly. Inflammation-derived fibrosis is usually accompanied by epithelial–mesenchymal transition (EMT) in epithelial cells. However, the precise molecular mechanism underlying mastitis-induced fibrosis remains unclear. Nuclear factor kappa-B (NF-κB) and Snail are key regulators of EMT. In this study, primary goat mammary epithelial cells (GMECs) were treated with 10 μg/mL lipopolysaccharide (LPS) for 14 d to mimic the in vivo mastitis environment. After LPS treatment, the GMECs underwent mesenchymal morphological transformation and expressed mesenchymal cell markers. Snail expression was induced by LPS and was inhibited by suppression of the TLR4/NF-κB signaling pathway. Snail knockdown alleviated LPS-induced EMT and altered the expression of inflammatory cytokines. Finally, we found that the expression of key molecules of the TLR4/NF-κB/Snail signaling pathway was increased in mastitis tissues. These results suggest that Snail plays a vital role in LPS-induced EMT in GMECs and that the mechanism is dependent on the activation of the TLR4/NF-κB signaling pathway.

## 1. Introduction

Mastitis is one of the most common clinical diseases in dairy animals; it may result in decreased milk production or a decrease in the quality of milk produced, and it disturbs the health and welfare of animals [1]. The economic impact on farms is very significant. Long-term mastitis is a prerequisite for mammary fibrosis, which probably enhances the epithelial–mesenchymal transition (EMT) by inducing the expression of inflammatory factors. There are two causes of long-term mammary infections in ruminants: delayed diagnosis and treatment of clinical mastitis [2,3], and long-term infection caused by chronic subclinical mastitis [4]. The combined prevalence of clinical mastitis and chronic subclinical mastitis in dairy cows exceeds 30% in the dairy industry [5,6]. To date, the molecular mechanisms underlying the pathologies of mammary fibrosis remain unclear. Similarly, knowledge of molecular mechanisms that may help in mastitis prevention—to benefit animal welfare and the economy—is also lacking.

Inflammation leads to injury and EMT in the epithelial cells [7]. Epithelial cells deliver signals—such as TGF-β [8]—to interstitial tissue, promoting fibroblasts activation and initiating tissue regenerative responses and fibrosis [9]. Lipopolysaccharide (LPS) is an important component of the outer membrane of Gram-negative bacteria. LPS can cause parenchymal injury and partial EMT of epithelial cells in vivo and can induce EMT in various cells in vitro [10,11]. However, the role of LPS in EMT in mammary epithelial cells remains largely unknown. Upon binding to TLR4, LPS promotes the expression of inflammatory cytokines through the nuclear translocation of nuclear factor kappa-B (NF-κB) [12]. Since several inflammatory factors are involved in the process of EMT [13,14], it is generally believed that NF-κB plays an important role in the pathogenesis of fibrosis. Notably, in addition to the pro-EMT effects of NF-κB, Snail mediates sequence-specific interactions with the E-box sequence of the E-cadherin promoter to induce EMT [15]. Snail is usually expressed in inflammatory fibrosis and tumor tissues [12,16]. Moreover, NF-κB ensures the inductive function of Snail by stabilizing Snail expression and promoting tumor invasion in the inflammatory microenvironment [17].

To investigate mastitis and the molecular mechanisms involved in mammary fibrosis, we utilized LPS to mimic an inflammatory mammary environment for the induction of EMT in goat mammary epithelial cells (GMECs). This study established that LPS-induced EMT was mediated by the TLR4/NF-κB/Snail signaling pathway, suggesting potential roles of Snail in goat mastitis.

## 2. Materials and Methods

### 2.1. Tissue Collection

Normal and mastitic goat mammary gland tissues were surgically isolated from three healthy and four mastitic Guanzhong dairy goats. All goats were two years of age and had begun lactating about 30 d prior. They were reared in Shaanxi Province (China). Mastitis was determined based on breast appearance, milk status and somatic cell count (SCC). Goats with SCC > 1 × 10^6^ were considered to be affected by mastitis [18,19,20]. Normal (uninfected) tissues were used to isolate primary GMECs, and both normal and mastitis-infected tissues were fixed in 4% paraformaldehyde after washing in phosphate-buffered saline (PBS) without Ca^2+^/Mg^2+^ to perform paraffin sections or immediately frozen in liquid nitrogen until protein and RNA extraction.

### 2.2. GMEC Isolation and Culture

Primary GMECs were obtained as previously described [21]. Briefly, under sterile conditions, mammary parenchymal tissue was obtained from the breast base of dairy goats, placed in PBS with penicillin and streptomycin at 100 U/mL and 100 µg/mL (Sigma-Aldrich, Co., St. Louis, MO, USA), respectively, then immediately transported to the laboratory. Tissue pieces were washed in PBS, minced to a size of 1 mm^3^, then placed into 24-well culture plates (one piece per well) and cultured in Dulbecco’s modified Eagle medium/nutrient mixture F-12 (DMEM/F-12, Gibco, Rockford, IL, USA) supplemented with 10% KnockOut Serum Replacement (Gibco) and 10 ng/mL epidermal growth factor (Sigma-Aldrich). The migration and growth of the primary cultured cells are shown in Figure A1. Validation results for the mammary epithelial cells are shown in Figure A2. The culture medium was replaced every 3 d until the cells around the tissue piece grew in layers. The cells were then passaged twice and frozen in liquid nitrogen. After recovery from the liquid nitrogen, cells were cultured in DMEM/F-12 containing 10% fetal bovine serum (FBS, Gibco) at 37 °C in a humidified atmosphere of 5% CO_2_ and passaged once for the experiment.

### 2.3. EMT Induction

GMECs were treated with either 10 μg/mL of LPS (055:B5, Sigma-Aldrich) or 2.5 ng/mL TGF-β1 (Invitrogen, Inc., Carlsbad, CA, USA), and the medium was changed every 2 d for 14 d to induce EMT. Cells were imaged under a microscope (Olympus, Co., Tokyo, Japan) every 24 h. When the degree of cell confluence reached 80% in the treatment process, the GMECs were passaged at a ratio of 1:6.

### 2.4. TAK-242 and QNZ Treatment

To inhibit the TLR4/NF-κB signaling pathway, GMECs were treated with TAK-242 (TLR4 inhibitor, AbMole, Co., Houston, TX, USA) or QNZ (NF-κB inhibitor, AbMole) for 6 h. The samples were collected after an inhibitor treatment for 3 h and LPS and an inhibitor treatment for 6 h. Cells were treated with LPS and an inhibitor for 14 d to attenuate the effects of LPS on EMT.

### 2.5. RNA Preparation and Polymerase Chain Reaction (PCR) Analysis

Total RNA was isolated using RNAiso Plus (Takara, Co., Ltd., Beijing, China), and reverse transcription was performed using a PrimeScript RT reagent Kit (Takara) according to the manufacturer’s protocols. PCR reactions were performed using a Fast SYBR Green Master Mix (Genstar, Co., Ltd., Beijing, China), and data collection and analysis were performed using a QuantStudio 6 Flex PCR system (Life Technologies, Co., Ltd., Carlsbad, CA, USA) with GraphPad Prism 6 software. Table A1 shows the sequences of specific PCR primers purchased from Sangong Biotech (Sangong Biotech, Co., Ltd., Shanghai, China). Relative mRNA expression was normalized to the level of GAPDH mRNA in the same sample.

### 2.6. Protein Extraction and Western Blotting Analysis

The GMECs and mammary tissues were lysed using high-efficiency RIPA lysate (Solarbio, Co., Ltd., Beijing, China) or a nuclear protein extraction kit (BestBio, Co., Ltd., Nanjing, China) to obtain total proteins, nuclear proteins and cytoplasmic proteins. The concentration of protein samples was determined using a BCA kit (Genstar). For Western blotting, samples with a loading buffer (CWBIO, Co., Ltd., Beijing, China) were separated using sodium dodecyl sulfate-polyacrylamide gel electrophoresis on gels (Jingcai Biological, Co., Ltd., Xi’an, China) and blocked in Tris-buffered saline Tween (TBST) with 5% non-fat milk (BBI, Co., Ltd., Shanghai, China) for 2 h at 25 °C after being transferred to a polyvinylidene fluoride membrane (Millipore, Sigma-Aldrich). The membranes were then incubated with specific primary antibodies at 4 °C overnight and with horseradish peroxidase-conjugated secondary antibodies (1:4000, diluted in TBST) at 25 °C for 2 h. The dilution ratios of the primary antibodies are listed in Table A2. After antibody incubation, the membrane was washed with TBST and photographed using a G:BOX Chemi XRQ (Syngene, Co., Frederick, MD, USA) imaging system.

### 2.7. Immunofluorescence Staining

The GMECs were treated with either LPS or inhibitors, as described above. GMECs were washed three times with PBS, fixed in 4% paraformaldehyde for 10 min, and permeabilized in 0.2% Triton X-100 (Sigma-Aldrich) for 5 min at 25 °C. After permeabilization, the cells were blocked for 2 h at 25 °C in PBS containing 1% FBS after washing in PBS. The cells were then incubated with specific primary antibodies overnight at 4 °C and incubated with the secondary antibody (1:100, diluted in PBS containing 1% FBS) for 2 h at 25 °C in the dark; the dilution ratio of antibodies is shown in Table A2. The cells were then incubated for 5 min with Hoechst 33,342 (1:1000, diluted in PBS containing 1% FBS, Sigma) after washing. Finally, the cells were viewed and photographed using an inverted fluorescence microscope (Zeiss, Co., Oberkochen, Germany).

### 2.8. Immunohistochemistry

Paraffin sections were deparaffinized, hydrated in a graded ethanol series, and then boiled for 15 min in citrate antigen retrieval solution (Solarbio). The sections were then stained using a universal streptavidin-perosidase staining kit (Solarbio) as previously described [22]. Briefly, the sections were treated with 0.3% H_2_O_2_ for 30 min at 37 °C to quench endogenous peroxidase activity, washed with PBS, and blocked with pre-immune serum for 30 min at 37 °C. Following the manufacturer’s instructions, the sections were incubated with an anti-Snail (1:100, diluted in PBS, Santa Cruz Biotechnology, Inc., Dallas, Texas, USA, sc-393172) specific primary antibody at 4 °C. The secondary antibody was incubated at 37 °C for 1 h and then incubated with streptavidin–biotin peroxidase for 30 min at 25 °C. Finally, the sections were visualized with diaminobenzidine lightly counterstained with hematoxylin, dehydrated, and mounted with coverslips. Sections were viewed under a microscope (Nikon, Co., Tokyo, Japan) after drying at 25 °C.

### 2.9. siRNA Transfection

The GMECs were treated with LPS for 14 d and then transfected at a 50% confluence with negative control (NC) siRNA or specific siRNA of Snail (GenePharma, Co., Ltd., Shanghai, China) using TurboFect transfection reagent (Thermo, Rockford, IL, USA). After 12 h of transfection, the culture medium was replaced with the LPS-treated medium. The silencing efficiency of three different siRNAs was evaluated using real-time PCR after transfection for 24 h and Western blotting for 72 h. The Snail siRNA sequence was F-GCUCUUUCCUCGUCAGGAATT, R-UUCCUGACGAGGAAAGAGCTT.

### 2.10. Statistical Analysis

The Shapiro–Wilk test was used to assess normality, and statistical analysis was performed when significance values exceeded 0.05. Data are presented as means ± standard deviation. Statistical significance was set at *p* < 0.05. All data were representative of at least three different experiments and were statistically analyzed using SPSS software (version 20.0, IBM Corp., Armonk, NY, USA). Student’s *t*-test was used for comparisons between the two groups. Statistical differences among different groups (>2) were evaluated using a one-way ANOVA with multiple comparisons among groups tested using Tukey’s *post hoc* test.

## 3. Results

### 3.1. LPS Induces EMT in GMECs

In this study, the conditions of LPS treatment were based on previously reported methods [23]. As shown in Figure A3, LPS induced inflammatory reactions in the GMECs by assessing the expression of inflammation-related factors. As TGF-β signaling has been shown to play an important role in EMT [24], we used TGF-β1 as a positive control to test whether LPS induced EMT in GMECs. All the cells were cultured for 14 d and passaged according to the methods described in Section 2.2. As shown in Figure 1A, the cell morphology changed from epithelial-like to mesenchymal-like after 14 d of treatment with 2.5 ng/mL TGF-β1 or 10 μg/mL LPS. The immunofluorescence results revealed that treatment with LPS or TGF-β1 led to low expression levels of E-cadherin and high expression of N-cadherin and vimentin (Figure 1B–D). In accordance with the immunofluorescence results, the protein expression of E-cadherin was markedly downregulated and levels of N-cadherin, vimentin and collagen III were markedly upregulated compared with the control (*p* < 0.05; Figure 1E). In addition, we observed that β-casein expression decreased in a time-dependent manner, and alpha-smooth muscle actin (α-SMA) showed the opposite trend (Figure A4).

### 3.2. NF-κB/Snail Signaling Pathway Mediates LPS-Induced EMT in GMECs

p65 and Snail (Snail1) mRNA exhibited higher expression after LPS challenge (*p* < 0.05), but Slug (Snail2) mRNA showed no difference compared with the control cells (*p* > 0.05). The protein expression of phosphorylated p65 (p-p65) and Snail also improved (*p* < 0.05; Figure 2A).

We assessed the expression patterns of p65 and Snail. As shown in Figure 2C, the levels of p-p65 were increased from the second day to the 16th day after LPS challenge compared to the control (*p* < 0.05; Figure 2C). Unlike p-p65, Snail was markedly upregulated on the sixth day after treatment (*p* < 0.05; Figure 2B), and its protein expression began on the 10th day (Figure 2C). To further validate whether NF-κB/Snail was involved in the process of LPS-induced EMT, we used either TAK-242 or QNZ to inhibit the TLR4/NF-κB pathway, and 5 μM and 5 nM working concentrations of TAK-242 and QNZ, respectively, were screened (Figure A5). After the treatment of TAK-242 or QNZ, the elevated levels of Snail and p-p65 induced by LPS were significantly suppressed (*p* < 0.05; Figure 2D).

### 3.3. p65 Nuclear Translocation Is Required for LPS-Induced Snail Expression

In this study, nuclear and cytoplasmic proteins were isolated to analyze the expression of NF-κB and Snail. Western blotting analysis showed that, compared with the distribution of p65 in the cytoplasm (average of relative values = 80.8%) and nucleus (19.2%) of the control group, LPS promoted the transfer of p65 from the cytoplasm (25.7%) to the nucleus (74.3%; Figure 3A,B). Meanwhile, the trend of Snail expression in total protein or nuclear protein was the same as that of p-p65 or p65 (Figure 3A). Moreover, there was no remarkable difference between the expression levels of Snail in the total protein and nuclear protein samples (*p* > 0.05; Figure 3C). Immunofluorescence staining revealed that p65 was mainly expressed in the cytoplasm of the control cells but was expressed in the nucleus of the LPS group (Figure 3D). In line with the Western blotting results, Snail was not expressed in the control cells, but was observed in the LPS group (Figure 3D). Furthermore, the inhibition of the TLR4/NF-κB signaling pathway reduced p65 nuclear translocation and the nuclear expression of Snail (*p* < 0.05; Figure 3E).

We further investigated the effect of the inhibition of the TLR4/NF-κB signaling pathway on EMT in LPS-treated cells. As expected, the inhibition of TLR4/NF-κB led to the attenuation of LPS-induced EMT. The protein expression of E-cadherin was increased, whereas the expression levels of N-cadherin, vimentin and collagen III were markedly decreased compared to those of the LPS group (*p* < 0.05; Figure 3F).

### 3.4. Knockdown of Snail Attenuates LPS-Induced EMT

To address whether Snail plays a role in governing LPS-induced inflammation and EMT, we transfected siRNA to knock down Snail mRNA expression. As shown in Figure 4A,B, Snail–siRNA transfection resulted in no significant changes in the p65 mRNA levels or p-p65 (*p* > 0.05). Thereafter, we investigated whether the inflammatory cytokines were regulated by Snail. Notably, compared with the NC group, Snail knockdown effectively reduced the mRNA expression of inflammatory cytokines, including IL-6 and TGF-β1. However, the expression of IL-1β, TNF-α and IL-8 mRNA was increased (*p* < 0.05; Figure 4C). In addition, the GMECs acquired E-cadherin expression and exhibited a decrease in the expression of N-cadherin, vimentin and collagen III after Snail knockdown (*p* < 0.05; Figure 4D).

### 3.5. Snail Is Expressed in Mastitis Tissue

To confirm the role of the TLR4/NF-κB/Snail signaling pathway in mastitis in vivo, the expression levels of the critical molecules of TLR4/NF-κB signaling and the transcriptional factors of EMT were detected. Compared with the normal group, the mastitis group showed significant upregulation of the mRNA levels of the LPS-mediated pathway-related genes, TLR4 and MyD88, as well as increased levels of inflammatory cytokines, TNF-α, IL-8 and TGF-β1; the expression of p65 and Snail mRNA were also increased (*p* < 0.05; Figure 5A,B). In the normal mammary gland, the structure of the alveolar lumen was complete, and the cavity was swollen with milk. In mastitis, a histopathological examination showed a massive recruitment of inflammatory cells into the alveolar lumen. In the normal group, connective tissue was closely connected with acini, whereas in mastitis, the fibers in the connective tissue were loosely connected. Snail was primarily detected in the cytoplasm of the mammary parenchymal cells, but not in the connective cells (Figure 5C). In addition, Western blotting analysis showed that the p-p65 and Snail expression levels were upregulated in the mastitis tissues compared to those of the normal tissues (Figure 5D).

## 4. Discussion

This study elucidated the role of the NF-κB/Snail signaling pathway in EMT in mastitis. Based on in vitro and in vivo experiments, we found that LPS promoted Snail expression through the TLR4/NF-κB inflammatory signaling pathway, leading to EMT in the GMECs.

In this study, a mesenchymal-like transformation was not observed until 7 d after the LPS induction. However, other cells showed significant morphological changes after 24–72 h of LPS treatment [25,26]. This result is related to the levels of inflammatory factors secreted by the mammary epithelial cells. According to previous reports, the secretion levels of TNF-α and IL-6 in mammary epithelial cells are significantly lower than in alveolar epithelial cells after treatment with the same concentration of LPS [27,28,29]. In addition to the changes in cell morphology, our data showed that LPS increased the expression of Snail. Interestingly, the results of this study showed that Snail was barely expressed in the control cells, though it was expressed both inside and outside the nucleus in the LPS group. Snail nuclear translocation is an essential mechanism for the transcriptional inhibition of E-cadherin; however, we did not obtain images of the moment of the nuclear translocation of Snail because Snail protein is unstable and is largely regulated by the ubiquitination–proteasome pathway [30,31]. In addition, the subcellular localization of Snail in GMECs may be affected by such post-translational modifications as phosphorylation and lysine oxidation [32].

Snail has been previously reported to aggravate or suppress inflammatory responses and to promote fibrosis [16,33]. In this study, Snail knockdown inhibited the mRNA expression of IL-6 and TGF-β1. TGF-β is first secreted in the early stages of inflammation to counter the inflammatory response; however, in the case of chronic injury, it becomes the most effective inducer of Snail [34]. Chen et al. recently reported that an in vitro supplementation of a sufficient dose of TGF-β induced EMT in bovine mammary epithelial cells [35]. TGF-β also inhibits IL-6 secretion and promotes fibrotic EMT by recruiting Smad proteins to Snail [36,37]. These findings support our results, suggesting that there is a mechanism of positive-feedback regulation between Snail, IL-6 and TGF-β1, and that Snail may play an anti-inflammatory and pro-fibrosis role in the early and late periods of mastitis, respectively. In addition, Snail knockdown significantly promoted the mRNA expression of IL-1β, TNF-α and IL-8, indicating that Snail may have an antagonistic effect on certain inflammatory factors. Similarly, Snail is involved in the inflammatory inhibition mediated by myeloid Notch1 signaling [33]. Combined with the positive feedback mechanism of Snail, we speculate that the primary role of Snail in mastitis is to inhibit inflammation. In the in vivo detection of the expression of TLR4/NF-κB/Snail, we found that the critical members of this signaling pathway were upregulated in all the mastitis glands. However, the upregulation of p-p65 does not necessarily lead to the expression of the Snail protein, corroborating our initial understanding. The underlying mechanism of this is consistent with Snail degradation and inflammation severity [38,39]. In addition, our data showed that long-term LPS treatment significantly increased α-SMA expression, suggesting that EMT could be sustained [40]. Therefore, based on the phased role of TGF-β/Snail in mastitis, further tests are needed to evaluate the appropriate intervention time point and reversibility of EMT if Snail is to be used as a target for the treatment of mastitis.

Another important implication of this study is that Snail expression was detected only in the mammary parenchyma of mastitis, which clearly distinguishes type II from type III EMT. Type II EMT is attributed to abnormal wound healing and fibrotic disease, while type III EMT is associated with tumor cell migration and invasion. However, inflammation, type II EMT and type III EMT are usually correlated to a certain extent, and type II EMT can induce and enhance type III EMT [41,42]. Therefore, fibrotic diseases are mostly associated with the poor prognosis of tumors [43,44]. In this study, Snail was expressed in the parenchyma but not in the mesenchyme, indicating that a partial EMT occurred in epithelial cells. The research on fibrosis in other organs suggests that the transformation of adult epithelial cells has greater potential; they remain in the anatomical positions in which they originate. Although epithelial cells do not have the ability to enter the interstitial tissue, factors secreted by epithelial cells with partial EMT enter the interstitial tissue via paracrine signaling, further exacerbating fibrosis [45,46]. In contrast, Snail is expressed in the fibrotic tissues of goats but is rarely expressed in the normal cells. Thus, Snail expression can be used as a marker of goat mammary fibrosis. These findings suggest that Snail is a promising target for the treatment of goat mammary fibrosis.

In the field of human medicine, mammary fibrosis is a preclinical disease and a prognostic complication of breast cancer [47]. Thus, alternative studies in laboratory and large animal models are necessary and useful for testing therapies targeting Snail. Considering the rationality of the experimental design and the degree of similarity to human structures, goats have been used as a relatively suitable substitute for rodents in the research of osteoporosis, nerve injury and regenerative medicine [48,49,50]. Given that non-neoplastic goat mammary fibrosis is more common, goat tissues may have greater potential for translational medical research on human mammary fibrosis. In ruminants, the inhibition of the EMT of mammary epithelial cells by targeting Snail offers an attractive potential way to restore their normal production capacity.

## 5. Conclusions

This study provides evidence that LPS induces EMT in GMECs via NF-κB-dependent Snail expression, suggesting that the EMT of mammary epithelial cells may occur in mastitis. Understanding the mechanism of LPS-induced EMT in mammary epithelial cells is essential for identifying novel targets for the treatment of mastitis in high-yielding dairy animals.

## Figures and Tables

**Figure 1 animals-11-03422-f001:**
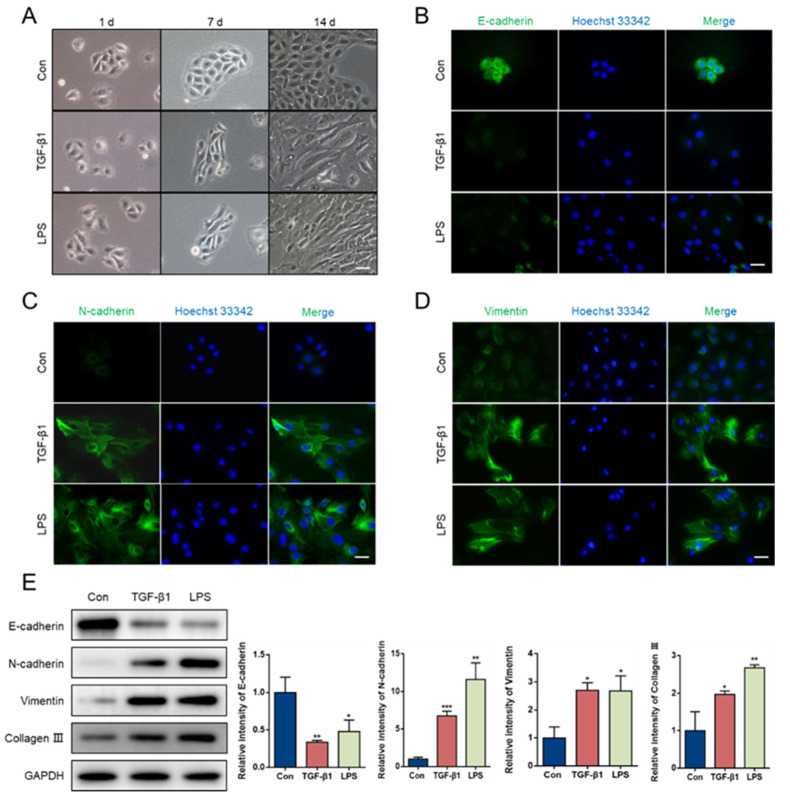
LPS induced EMT in GMECs. (**A**) Changes of cell morphology during LPS (10 μg/mL) or TGF-β1 (2.5 ng/mL) treatment. Immunofluorescence staining of E-cadherin (**B**), N-cadherin (**C**) and Vimentin (**D**) in GMECs were observed after LPS or TGF-β1 treatment for 14 d. (**E**) Western blotting results of EMT markers. (original western blot figures in Appendix A). * Indicates a significant difference compared with the control group, * *p* < 0.05, ** *p* <0.01, *** *p* <0.001. Scale bar = 50 μm.

**Figure 2 animals-11-03422-f002:**
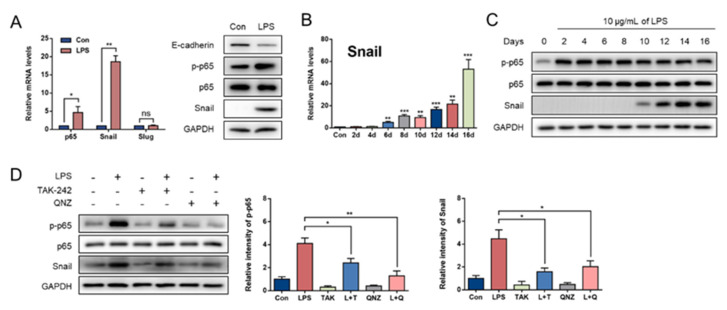
LPS activated NF-κB/Snail signaling pathway in GMECs. (**A**) mRNA expression of p65 and Snail after LPS treatment for 14 d. (original western blot figures in Appendix A). (**B**) mRNA expression of Snail during LPS treatment for 16 d. * Indicates a significant difference compared with the control group. * *p* < 0.05, ** *p* < 0.01, *** *p* < 0.001. (**C**) Protein expression of p65 and Snail during LPS treatment for 16 d. (original western blot figures in Appendix A). (**D**) The effects on p65/Snail expression by inhibiting the TLR4/NF-κB signaling pathway with TAK-242 (TAK/T, TLR4 inhibitor) or QNZ (Q, NF-κB inhibitor), L + T = LPS + TAK-242, L + Q = LPS + QNZ. (original western blot figures in Appendix A). * Indicates a significant difference between two groups, * *p* < 0.05, ** *p* <0.01, *** *p* < 0.001, ns = no significance.

**Figure 3 animals-11-03422-f003:**
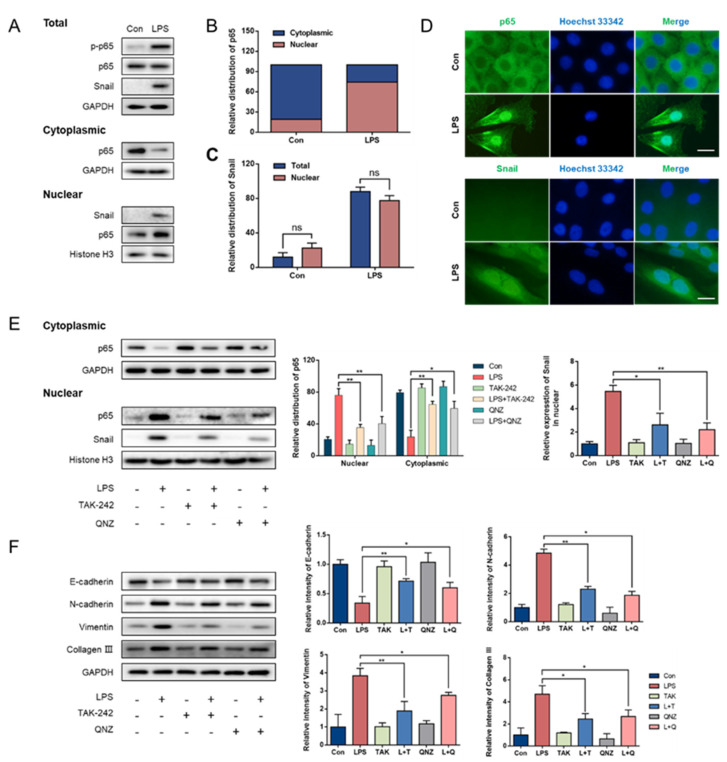
Snail expression required p65 nuclear translocation in GMECs. (**A**) Expression of p65 and Snail inside and outside the nucleus after LPS treatment for 14 d. (original western blot figures in Appendix A). (**B**) Relative distribution of p65 in nuclear proteins and cytoplasmic proteins. (**C**) Relative distribution of Snail in nuclear proteins and total proteins. (**D**) Immunofluorescence staining showed the localization of p65 and Snail. (**E**) p65 and Snail protein expression inside and outside the nucleus after inhibition of the TLR4/NF-κB signaling pathway. (original western blot figures in Appendix A). (**F**) Protein expression of EMT markers after TLR4/NF-κB inhibition. (original western blot figures in Appendix A). * Indicates a significant difference between two groups, * *p* < 0.05, ** *p* < 0.01. Scale bar = 20 μm.

**Figure 4 animals-11-03422-f004:**
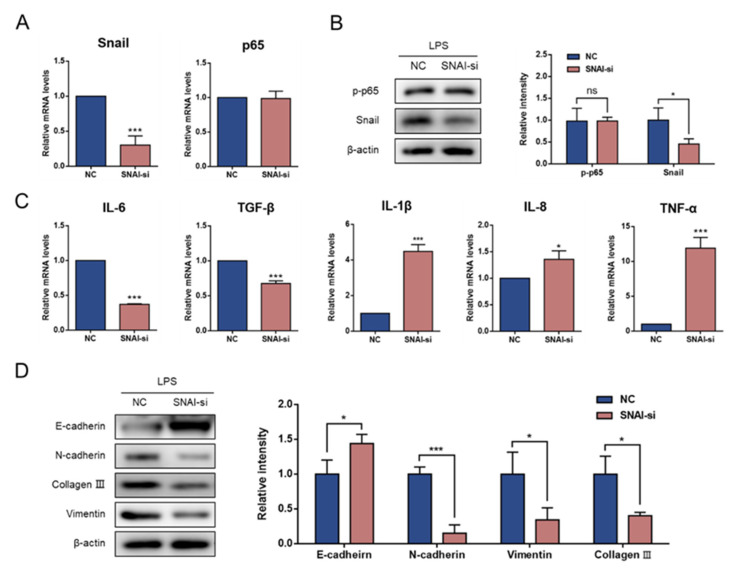
Snail knockdown changed the expression of inflammatory cytokines and EMT markers. Snail–siRNA was transfected into GMECs after LPS treatment for 14 d. (**A**) mRNA expression levels of p65 and Snail after Snail knockdown. (**B**) Western blotting results of p65 and Snail after Snail knockdown. (original western blot figures in Appendix A). (**C**) mRNA expression levels of inflammatory cytokines after Snail knockdown. (**D**) Western blotting results of EMT markers. (original western blot figures in Appendix A). * Indicates a significant difference compared with the NC group, * *p* < 0.05, *** *p* < 0.001.

**Figure 5 animals-11-03422-f005:**
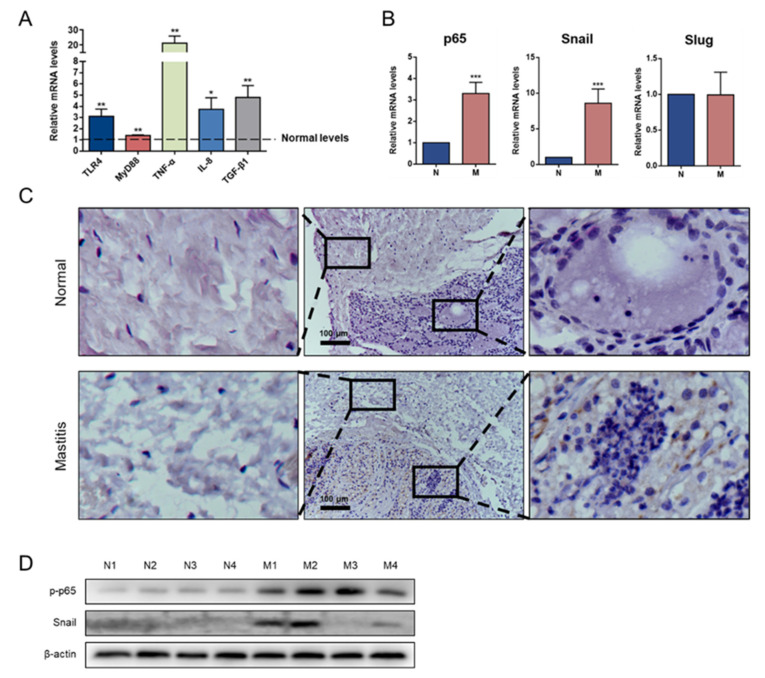
Snail was expressed in goat mastitic tissues. (**A**) mRNA expression levels of TLR4 pathway members and inflammatory cytokines. Normal levels represent the gene expression levels of normal goats. (**B**) mRNA expression levels of p65 and Snail. (**C**) Immunohistochemistry staining showed the localization of Snail in normal (N) and mastitis (M) mammary gland tissues. (**D**) Protein expression levels of p-p65 and Snail in N/M mammary gland tissues. N3 and N4 samples were obtained from the contralateral breast of the same dairy goat. * Indicates a significant difference compared with the normal goats, (original western blot figures in Appendix A). * *p* < 0.05, ** *p* < 0.01, *** *p* < 0.001. Scale bar = 100 μm.

## Data Availability

The data that support the findings of this study are available from the corresponding author upon reasonable request.

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
