# Peer review of "NF-κB–Dependent Snail Expression Promotes Epithelial–Mesenchymal Transition in Mastitis"

_animals, 2021, doi:10.3390/ani11123422_

Round 1
Reviewer 1 Report
The contents of this study are very interesting contents, clearly explained and represented by means of graph and images. However, few minor revision are still needed.
In the manuscript are often used grammatical forms like “in this study we demonstrated”. I suggest to use impersonal forms like “This study shows”
Line 30: I suggest correct “d” with “Days” since it’s the first time that you use it in the paper. In the rest of the manuscript the authors can use the abbreviation.
Line 45: there is a double spacing. Please correct it.
Line 67 there is a typo. I suggest to correct “mastitis” with “mastitic” of “affected by mastitis”
Line 69 Please add the limit of somatic cell count used to consider the udder affected by mastitis.
Line 70 Please add appropriate reference for the tissue isolation method.
Line 75 Please add appropriate references for all the procedure described
Line 94 Please add appropriate references for all the procedure described
Line 101 Please add appropriate references for all the procedure described
Line 132 Please add appropriate references for all the procedure described
Lines 66-68: Please explain how the surgery was performed
Sub-section 2.10: Please assess the normality and, if needed, correct the analysis accordingly
Lines 41- 42: The authors should mention the impact of animal welfare to clarify the importance of the mastitis.
Section 4 - Discussion: I suggest discussing the potential of using the goat as an animal model translational medicine
Section 4 – Discussion: This section could be enhanced comparing the author’s results with the data of similar studies performed in goat or other ruminants.
Section 4 – Discussion: The authors should specify when they are comparing their results with references obtained in humans or other species
Line 347-348: The author should briefly describe, here or in the discussion, if there are any treatments for mastitis based on the mechanism examined in veterinary or human medicin
In the bibliography some journal’s names seems to not be abbreviated. Please correct
Author Response
Response to Reviewer 1 Comments
Point 1: In the manuscript are often used grammatical forms like “in this study we demonstrated”. I suggest to use impersonal forms like “This study shows”.
Response 1: Checked & revised.
Point 2: Line 30: I suggest correct “d” with “Days” since it’s the first time that you use it in the paper. In the rest of the manuscript the authors can use the abbreviation.
Response 2: Checked & revised.
Point 3: Line 45: there is a double spacing. Please correct it.
Response 3: Checked & revised.
Point 4: Line 67 there is a typo. I suggest to correct “mastitis” with “mastitic” of “affected by mastitis”.
Response 4: Checked & revised.
Point 5: Line 69 Please add the limit of somatic cell count used to consider the udder affected by mastitis.
Response 5: Checked & revised.
Point 6: Line 70 Please add appropriate reference for the tissue isolation method.
Line 75 Please add appropriate references for all the procedure described.
Line 94 Please add appropriate references for all the procedure described.
Line 101 Please add appropriate references for all the procedure described.
Line 132 Please add appropriate references for all the procedure described.
Response 6: Checked & revised. We have added references where appropriate.
Point 7: Lines 66-68: Please explain how the surgery was performed.
Response 7: Checked & revised.
Point 8: Sub-section 2.10: Please assess the normality and, if needed, correct the analysis accordingly.
Response 8: Checked & revised.
Point 9: Lines 41- 42: The authors should mention the impact of animal welfare to clarify the importance of the mastitis.
Response 9: Checked & revised.
Point 10: Section 4 - Discussion: I suggest discussing the potential of using the goat as an animal model translational medicine.
Response 10: Checked & revised.
Point 11: Section 4 – Discussion: This section could be enhanced comparing the author’s results with the data of similar studies performed in goat or other ruminants.
Response 11: Checked & revised. Because there are few reports about mastitic EMT in ruminants, we only added one reference about cow.
Point 12: Section 4 – Discussion: The authors should specify when they are comparing their results with references obtained in humans or other species.
Response 12: Checked & revised. We have compared the results of this study more clearly with reported data.
Point 13: Line 347-348: The author should briefly describe, here or in the discussion, if there are any treatments for mastitis based on the mechanism examined in veterinary or human medicine.
Response 13: Checked & revised.
Point 14: In the bibliography some journal’s names seems to not be abbreviated. Please correct.
Response 14: Checked & revised.

Reviewer 2 Report
Introduction. It will be nice if the authors could include a few more points in the introduction of the manuscript, possibly with a new paragraph.
The objectives of the work should be stated very clearly.
Table 1. Please move to supplementary material.
Sub-sections 2.5 and 2.6. All the details about the materials should be moved to supplementary material.
2.10. Please provide evidence about normal distribution of the results. Otherwise, please modify the analysis using non-parametric techniques.
Figure 1, E. Please colourise the histograms.
Figure 2, A, B, D. Please colourise the histograms.
Figure 3, B, E, F. Please colourise the histograms.
Figure 4, A, B, C, D. Please colourise the histograms.
Figure 5, A, B. Please colourise the histograms.
The results are difficult to read in the form of text. Please include tables to present the results, which will make reading and understanding easier for future readers.
The discussion covers all the topics, but it is rather shallow and does not go into great depth and into the heart of the matter. It should be significantly extended, to become at least double and really present all the prospects of this work. Moreover, some recent significant references are missing and must be added.
The manuscript must be corrected as indicated and resubmitted in a better form.
The manuscript is written in poor English language; therefore, it should be corrected by a native speaker. As it is, the manuscript is not publishable due to its serious language flaws.
Author Response
Response to Reviewer 2 Comments
Point 1: Introduction. It will be nice if the authors could include a few more points in the introduction of the manuscript, possibly with a new paragraph.
Response 1: Checked & revised.
Point 2: The objectives of the work should be stated very clearly.
Response 2: Checked & revised. We have further cleared the objectives of this work in the Introduction and Discussion sections.
Point 3: Table 1. Please move to supplementary material.
Response 3: Checked & revised.
Point 4: Sub-sections 2.5 and 2.6. All the details about the materials should be moved to supplementary material.
Response 4: Checked & revised. We have moved these details to Appendix A.
Point 5: 2.10. Please provide evidence about normal distribution of the results. Otherwise, please modify the analysis using non-parametric techniques.
Response 5: Checked & revised.
Point 6: Figure 1, E. Please colourise the histograms.
Figure 2, A, B, D. Please colourise the histograms.
Figure 3, B, E, F. Please colourise the histograms.
Figure 4, A, B, C, D. Please colourise the histograms.
Figure 5, A, B. Please colourise the histograms.
Response 6: Checked & revised. In addition to colourize the histograms in the text, we also colourize the histograms in Appendix A.
Point 7: The results are difficult to read in the form of text. Please include tables to present the results, which will make reading and understanding easier for future readers.
Response 7: Checked & retained. Thank you for your valuable advice. We tried to use tables to present the data, but this might make it harder to read. Therefore, we still retained the presentation form of the original data. In addition, we invited native English speakers to revise the manuscript. We believe that the description of the results is now easier for readers to read and understand.
Point 8: The discussion covers all the topics, but it is rather shallow and does not go into great depth and into the heart of the matter. It should be significantly extended, to become at least double and really present all the prospects of this work. Moreover, some recent significant references are missing and must be added.
Response 8: Checked & revised.
Point 9: The manuscript is written in poor English language; therefore, it should be corrected by a native speaker. As it is, the manuscript is not publishable due to its serious language flaws.
Response 9: Checked & revised.

Round 2
Reviewer 2 Report
The authors have improved the manuscript by making the changes requested.
However, the language problems still persist and the manuscript is difficult to read.
Please consult a professional expert in English language for making the necessary corrections throughout the manuscript.
Hence, the manuscript cannot be accepted now, but serious revision in language is necessary.
Author Response
Response to Reviewer 2 Comments
Point 1: The authors have improved the manuscript by making the changes requested.
However, the language problems still persist and the manuscript is difficult to read.
Please consult a professional expert in English language for making the necessary corrections throughout the manuscript.
Hence, the manuscript cannot be accepted now, but serious revision in language is necessary.
Response 1: We are sorry to trouble you with the language problem.
We have carefully revised the manuscript language.